# New Perspective for Macroalgae-Based Animal Feeding in the Context of Challenging Sustainable Food Production

**DOI:** 10.3390/plants12203609

**Published:** 2023-10-18

**Authors:** Georgia M. González-Meza, Joel H. Elizondo-Luevano, Sara P. Cuellar-Bermudez, Juan Eduardo Sosa-Hernández, Hafiz M. N. Iqbal, Elda M. Melchor-Martínez, Roberto Parra-Saldívar

**Affiliations:** 1Tecnologico de Monterrey, Institute of Advanced Materials for Sustainable Manufacturing, Monterrey 64849, Mexico; georgia.gonzalez@tec.mx (G.M.G.-M.); joel.elizondolv@uanl.edu.mx (J.H.E.-L.); eduardo.sosa@tec.mx (J.E.S.-H.); hafiz.iqbal@tec.mx (H.M.N.I.); 2Tecnologico de Monterrey, School of Engineering and Sciences, Monterrey 64849, Mexico

**Keywords:** animal feeding, animal nutrition, bioactive compounds, Chlorophyceae, macroalgae, Phaeophyceae, Rhodophyceae, seaweed

## Abstract

Food production is facing challenging times due to the pandemic, and climate change. With production expected to double by 2050, there is a need for a new paradigm in sustainable animal feed supply. Seaweeds offer a highly valuable opportunity in this regard. Seaweeds are classified into three categories: brown (Phaeophyceae), red (Rhodophyceae), and green (Chlorophyceae). While they have traditionally been used in aquafeed, their demand in the feed market is growing, parallelly increasing according to the food demand. Additionally, seaweeds are being promoted for their nutritional benefits, which contribute to the health, growth, and performance of animals intended for human consumption. Moreover, seaweeds contain biologically active compounds such as polyunsaturated fatty acids, antioxidants (polyphenols), and pigments (chlorophylls and carotenoids), which possess beneficial properties, including antibacterial, antifungal, antiviral, antioxidant, and anti-inflammatory effects and act as prebiotics. This review offers a new perspective on the valorization of macroalgae biomass due to their nutritional profile and bioactive components, which have the potential to play a crucial role in animal growth and making possible new sources of healthy food ingredients.

## 1. Introduction

Macroalgae, also known as seaweed, offers a novel and value-added dietary ingredient in formulated diets for animal feeding [1]. In applied phycology, the term macroalgae usually includes macroscopic algae sensu stricto, where the cell structure is eukaryotic; these organisms are capable of photosynthesis and exist in marine and freshwater environments [2]. Macroalgae are organisms found in the ocean and water bodies, encompassing 10,000 species. They are categorized into three different groups depending on their pigmentation: brown seaweed (Phaeophyceae), red seaweed (Rhodophyceae), and green seaweed (Chlorophyceae) [3].

Seaweed proliferates and is found in coastal and estuarine areas [3]. Brown seaweeds are characterized by their long, thick, and leathery appearance, with some species capable of reaching lengths of up to 45 m. In contrast, red seaweeds are relatively small and can vary in color from red to purple or brownish-red, with a maximum size of around 1 m. Green seaweeds and red seaweeds share similar sizes and have a close relationship with each other [4].

Macroalgae exhibit a diverse nutritional composition, including carbohydrates, proteins, lipids, vitamins, and minerals (e.g., iodine, iron, calcium) [5]. Furthermore, macroalgae are rich in bioactive compounds such as omega-3 fatty acids, carotenoids, phycocyanin, and polysaccharides. These compounds possess antimicrobial, antiviral, anti-inflammatory, antioxidant, prebiotic, and even anticancer properties, making them crucial for various physiological functions in animals [6,7,8]. Macroalgae contain unique carbohydrates, such as laminarin and fucoidan, which can function as prebiotics, promoting the growth of beneficial gut bacteria and enhancing nutrient digestion and absorption in mammals [9,10]. Furthermore, bioactive compounds in macroalgae can help reduce methane production in ruminants and improve digestibility [11]. Including macroalgae in animal diets has been shown to improve nutrient utilization, support digestive health, and reduce gastrointestinal disorders [12].

Studies have demonstrated that incorporating macroalgae into fish and animal diets and human functional food can positively impact growth performance [13]. However, the potential of macroalgae is focused on its use as an animal feed resource rather than a direct food source for human consumption. The balanced nutrient profile of macroalgae supports optimal growth rates and body weight gain in various animal species [14,15]. Furthermore, the bioactive compounds in macroalgae, such as polyphenols, flavonoids, and antioxidants, contribute to improved immune function and overall health. These compounds exhibit antioxidant, antimicrobial, anti-inflammatory, and immunomodulatory properties [16,17,18]. By including macroalgae in animal diets, animals may experience reduced oxidative stress, enhanced immune response, and improved disease resistance [19,20].

Furthermore, utilizing macroalgae as a feed ingredient promotes environmental sustainability and offers a sustainable alternative to conventional feed ingredients like corn and soybeans, reducing competition among the food, feed, and biofuel industries. Also, macroalgae cultivation requires fewer resources (such as freshwater and arable land) than traditional feed crops [21,22]. Moreover, macroalgae cultivation can help mitigate nutrient pollution and eutrophication in aquatic environments by absorbing excess nutrients (e.g., nitrogen and phosphorus) from wastewater or aquaculture effluents [23,24]. Besides, macroalgae can capture and sequester carbon dioxide (CO_2_) during their growth [25,26]. Additionally, macroalgae supplementation in ruminant diets has shown promising results in reducing enteric methane (CH_4_) emissions, significantly contributing to greenhouse gas emissions [27]. For this reason, based on scientific evidence and research, macroalgae represents a sustainable and viable source of ingredients for animal feed production [1].

The purpose of this review is to summarize the properties of macroalgae as an animal feed source. Macroalgae are a promising source of biomass, providing nutrients and various products that are biologically, commercially, and nutritionally valuable [28,29]. We gathered information from scientific databases like PubMed, ResearchGate, and ScienceDirect. Our search included the keywords Macroalgae, seaweed, green seaweed, red seaweed, and brown seaweed, which we used as filters to search all fields. Moreover, we searched for the word composition, bioactive compounds, applications, and animal health benefits. Additionally, a comprehensive literature search was conducted for all periods using the genus names “Chlorophyceae”; “Rhodophyceae”; and “Phaeophyceae”. After searching, we carefully read the abstracts of the articles and selected relevant studies for review. Our eligibility criteria included original articles written in English or Spanish that discussed or utilized green, red, and brown seaweed.

## 2. Monitoring of Nutritional Profile and Bioactivities of Macroalgae

For centuries, macroalgae have been a valuable food source for humans and animals alike. These marine organisms are incredibly diverse [2]. Lately, there has been an increasing fascination with macroalgae’s nutritional and health advantages. This is primarily because they contain many vital nutrients and bioactive components. Nevertheless, it is only recently that their widespread cultivation and harvesting have become popular [30]. This article section examines the potential health advantages and nutritional value of macro and micronutrients from microalgae, contributing to their market expansion. Additionally, we highlight several crucial obstacles that still need to be addressed in this area. This section will discuss the importance of monitoring macroalgae’s nutritional profile and bioactivities and provide relevant bibliographic references.

Macroalgae display significant diversity in their nutritional composition, rendering them a valuable source of essential nutrients. They are abundant in proteins, vitamins (such as C, E, and B complex), minerals (calcium, iodine, iron, and magnesium), and dietary fiber [31]. Additionally, macroalgae contain various bioactive compounds like polysaccharides, polyphenols, carotenoids, and phycobiliproteins, contributing to their potential health benefits [32,33,34,35]. The bioactive compounds present in macroalgae possess several beneficial properties, making them promising candidates for functional foods and nutraceuticals; for instance, polysaccharides derived from macroalgae exhibit immunomodulatory, antioxidant, and antitumor activities [33,36]. Research has demonstrated that seaweed can benefit animal health, including antibacterial, antioxidant, and anti-inflammatory properties. This has been observed in various species, such as pigs, fish, chickens, and ruminants [28,37,38]. Seaweed’s chemical composition varies due to species, harvest time, and environmental conditions like temperature, light, salinity, and nutrients [39]. In addition, they can potentially improve gut health through prebiotic activity, which stimulates the growth of advantageous gut bacteria [9]. Additionally, the high content of polyphenols in certain macroalgae species contributes to their anti-inflammatory, antiviral, antibacterial, and cardioprotective effects [31,40,41]. Furthermore, certain species of macroalgae contain omega-3 fatty acids, which have the potential to offer benefits for cardiovascular health [42,43].

In contrast to macroalgae’s benefits in human and animal health, several reports have made consistent algae’s capacities for heavy metal and toxic compound accumulations. The marine ecosystems are exposed to organic and inorganic contaminants such as polycyclic aromatic hydrocarbons, organochlorine pesticides, and heavy metals [44]. Therefore, the monitoring of micropollutants in algae has demonstrated a crucial relationship between contaminant concentration and the macroalgae specie and morphology. Green algae, of the *Ulva* genus can accumulate naphthalene and benzo[a]pyrene at 68.57 and 56.14 ng g^−1^ concentrations, respectively. Due to its ability to accumulate these toxic substances, there is a risk of them entering the food chain, raising concerns for human health that should be studied and considered before its use [45]. In a red algae investigation, 72 food products containing Rhodophyta were analyzed for human consumption and this revealed a higher concentration of heavy metals such as Al, Cd, As, and I. Therefore, a continuous, mandatory evaluation and analysis of seaweed biomass destined for the food and feed industry is recommended [46].

In addition, seaweed aquaculture in the open ocean can deplete the dissolved oxygen because the algae could take nutrients from the medium in the sea. The proper range of dissolved oxygen for cultivation should be between 6.7 and 7.0 mg L^−1^, as a significantly lower concentration could adversely impact other ecosystems [47]. In consideration, it is essential to thoroughly evaluate where the farms will be located, considering the flow of nutrients and the distance from other coastal ecosystems.

Hence, extensive monitoring of macroalgae’s nutritional profile should be solved to ensure the safety of their consumption [29]. Macroalgae can accumulate harmful substances from the marine environment, posing a risk to human health [48]. Regularly monitoring the nutritional profile of macroalgae is essential to meet safety standards, as they contain bioactive compounds with health benefits [49]. For example, brown seaweeds have phlorotannins that offer antioxidant, anti-inflammatory, and anti-cancer benefits [33]. They also contain fucoidans, which have immunomodulatory, antitumor, and anticoagulant properties [50]. Therefore, monitoring the bioactivities of macroalgae is essential to identify new sources of bioactive compounds with potential health benefits. Table 1 lists the macromolecules commonly found in macroalgae, along with their corresponding reference.

Since the 1950s, seaweeds have been a crucial ingredient in many bio-stimulant products available in the market [69]. Seaweed is a nutritious source of vitamins, minerals, proteins, and dietary fibers. It also has anti-inflammatory and antioxidant benefits [70]. Seaweeds have both water-insoluble and -soluble fiber, which include cellulose, mannans, xylan, agars, alginic acid, furonan, laminarin, and porphyrin. These fibers are not only nutritious but also have potential uses in human consumption, such as seaweed-based meals, functional foods, and nutraceuticals [71,72]. Marine algae demonstrate promising potential as excellent sources of fiber, highlighting a wide range of chemical, physicochemical, and rheological characteristics that can offer nutritional benefits [71].

The polysaccharides present in seaweeds exhibit various beneficial properties such as anti-tumor and anti-herpetic activities, anticoagulant effects, LDL (low-density lipoprotein) cholesterol reduction in rats, antiviral properties, prevention of obesity, prevention of colorectal cancer, and prevention of diabetes [73,74]. Diverse types of seaweed contain varying amounts of polysaccharides, with the highest concentration found in species such as *Palmaria*, *Ascophyllum*, *Ulva*, and *Porphyra* [75]. These polysaccharides are not digested and act as dietary fibers, which can impact the digestibility of protein and minerals [76].

## 3. Biomass/Extracts Macroalgae for Animal Feeding Applications

The use of macroalgae or macroalgae extracts in animal feed has garnered significant attention due to the increasing demand for renewable and sustainable sources of animal protein, reducing the strain on land resources [77]. Numerous studies have investigated the incorporation of fresh or dried macroalgae and its extracts in feeding animals, focusing on aquatic organisms (Figure 1). Macroalgae metabolites have been found to enhance growth, boost immunity, reduce microbial load, and improve meat quality [12,78]. However, it is essential to note that macroalgae are primarily used as fortifiers in basal animal feed rather than as a whole feed source due to their essential amino acid content being considerably lower than that of traditional ingredients such as animal and soybean protein, fishmeal, and fish oil. Furthermore, macroalgae antinutrients can affect specific animal metabolisms, particularly in monogastric animals [79].

On the other hand, there is a growing need for alternatives to reduce or replace the indiscriminate use of antibiotics in animal diets. This need arises due to the detrimental effects of antibiotics, including the emergence of antibiotic-resistant bacterial strains, high residue concentrations in meat, and undesired alterations in gastrointestinal microbial communities. Consequently, many countries have banned the use of antibiotics as growth promoters [80]. Several macroalgae species with favorable antimicrobial activity have been identified as suitable candidates for inclusion in the diets of aquatic animals, cattle, rats, chickens, laying hens, and pigs [12]. Therefore, macroalgae can be considered a natural nutraceutical product that not only enhances the nutritional quality and meat production of feed but also enhances antioxidant activity, immunity, and overall animal health [20,81,82].

The composition of macroalgae metabolites can vary depending on factors such as species, geographic location, season, external conditions (pH, water temperature, sunlight intensity), and nutrient concentration in the water [79]. This variability provides ample opportunities to enhance feeding techniques by identifying ingredients with beneficial characteristics such as high nutritional profiles (amino acids, fatty acids, polysaccharides, vitamins, and minerals), digestibility, environmental and consumer safety, low production costs, year-round availability, and suitability as alternatives to fishmeal, animal protein, antibiotics, and immunostimulants [1,78].

### 3.1. Aquatic Organisms

Aquaculture has witnessed rapid growth in recent years to meet the escalating demand for seafood. However, the aquaculture industry’s utilization of over 70% of the world’s fishmeal, despite aquafeeds comprising only 4% of the total production of industrial feeds (which amounted to approximately 900–1000 million t in 2018), raises concerns about the long-term sustainability and impact on wild fish stocks due to its reliance on conventional feed ingredients including fish oil derived from wild-caught fish [83,84]. Biomass and extracts derived from macroalgae or seaweeds offer a promising alternative or replacement for these ingredients due to their high growth rates, abundance, and diverse chemical composition [85,86]. However, the nutritional composition of macroalgae can be variable, requiring careful selection and processing to optimize the inclusion level in their use in aquafeeds. Scientific investigations have focused on evaluating the impact of incorporating macroalgae or macroalgae extracts into aquafeed on growth performance, feed utilization, and health indicators. Macroalgae-derived feed supplements have positively impacted aquatic organisms’ growth, survival rates, digestibility, and immune responses in aquatic species [87]. The bioactive compounds in macroalgae, such as polysaccharides, polyphenols, and antioxidants, contribute to these beneficial effects by improving nutrient absorption, gut health, and immune function [1,88]. While macroalgae offer nutritional benefits, it is important to consider the presence of antinutrients in their biochemical composition when using them as feed [89]. Antinutrients are naturally occurring compounds that can interfere with nutrient absorption or utilization, potentially affecting animal health and performance [90]. Some common antinutrients in macroalgae include phytates, tannins, oxalates, and lectins [91]. Various processing methods can be employed to mitigate the adverse effects of antinutrients. These include heat treatments (e.g., blanching, steaming), fermentation, enzymatic treatments, and enzymatic liquefaction, which can reduce the levels of antinutrients and improve the overall nutritional value of macroalgae-based feeds [92,93,94]. On the other hand, the utilization of macroalgae extracts presents an alternative option as they are more concentrated, potentially offering higher concentrations of specific bioactive compounds such as antioxidants or immunostimulants [19,95,96]. This approach can effectively counteract the negative impacts of anti-nutrients in the macroalgae, ensuring their harmful effects are minimized.

The inclusion of macroalgae in aquafeeds can enhance the fatty acid profile of fish, increasing the levels of desirable omega-3 fatty acids. Legarda et al. (2021) conducted a study in which they examined the dietary incorporation of *Ulva fasciata* at three different inclusion levels (5%, 10%, and 20%) in the diets of *Seriola dorsalis* [97]. The study’s results indicated a significant increase in the levels of docosahexaenoic acid (DHA) in the muscle tissue of the fish. Sultana et al., in 2023, conducted a study to evaluate the effects of the dietary inclusion of the macroalgae *Hypnea* sp. on Nile Tilapia (*Oreochromis niloticus*). The study revealed that incorporating *Hypnea* sp. at a 10% inclusion level in the diets resulted in a significant increase in the levels of eicosapentaenoic acid (EPA) and docosahexaenoic acid (DHA) in the muscle tissue of the tilapia. This finding suggests that an appropriate inclusion of *Hypnea* sp. in aquafeed can serve as a crucial strategy to enhance the quality of meat in aquaculture [98]. Metabolites and bioactive compounds derived from seaweed have been evaluated for their beneficial effects on cultivated fish, shrimps, and oysters, including improved growth performance, enhanced digestibility, immunostimulatory and antioxidant activities, up-regulation of immune-related genes, resistance against viruses and bacteria, and tolerance to thermal and salinity stress [96,99,100,101,102,103,104,105,106,107]. Table 2 provides up-to-date information and references to significant research studies focusing on the effects of seaweeds and their extracts as bioactive ingredients in shrimp feeds. The discussed studies explore dosages and the effects on growth performance, gut health, antioxidant activity, gene regulation, immune responses, pathogen resistance, and stress in various shrimp species.

### 3.2. Poultry

Poultry is crucial in fulfilling the worldwide need for animal protein and supplying necessary nutrients to an ever-increasing population. However, the poultry industry faces numerous challenges, including the need for sustainable practices, disease management strategies, improved productivity, and the development of alternative feed ingredients [129]. Recent advancements in poultry nutrition have focused on alternative feed ingredients, including plant-based proteins and the incorporation of macroalgae extracts, which are alternative sources that offer potential benefits such as reduced environmental impact, improved animal health, enhanced meat quality, and meat shelf-life indicators [130,131]. Achieving optimal weight gain is a crucial objective for producers. Macroalgae is an alternative feed additive that can improve weight gain and promote sustainability. A study showed that supplementing broilers with 3% *Laminaria japonica* and cecropin led to better broiler growth performance and increased levels of antibodies against Newcastle disease [132]. Moreover, in the cecum, the growth of *Escherichia coli* was inhibited, while the growth of *Lactobacillus* was increased. Adding 2 to 3.5% of *Ulva* spp. to Boschveld chicken feed increased their weight gain and feed intake but did not affect the nutrient digestibility or feed utilization efficiency, according to a study by Nhlane et al. (2020) [133]. Balasubramanian et al. (2021) studied the addition of *Halymenia palmata* to broiler diets to enhance meat quality; they evaluated concentrations of 0.05%, 0.10%, 0.15%, and 0.25%. With increasing the inclusion levels of the red seaweed, the results indicated a linear decrease in the water-holding capacity, a crucial meat quality parameter. Additionally, incorporating red seaweed has proven to have a beneficial impact on broiler growth, nutrient absorption, microbial levels in feces, gas emissions from feces, blood composition, and tissue structure analysis [134]. On the other hand, macroalgae present a sustainable approach by offering natural compounds with antimicrobial characteristics, including tannins, which are polyphenolic compounds with antinutritional attributes [92]. Nonetheless, tannins have demonstrated antimicrobial, antioxidant, and anti-inflammatory properties, making them an intriguing prospect as bioactive agents to address the concerns associated with removing antimicrobials in the poultry sector [135]. Furthermore, according to Kulshreshtha et al. (2020), red algae-derived polysaccharides demonstrate antimicrobial properties due to their affinity for bacterial surface appendages, affecting *Salmonella enteritidis* virulence factors [136]. Dietary supplementation with macroalgae has been observed to increase the population of beneficial probiotic bacteria while reducing harmful enteric bacteria in poultry, as well as improving egg production and quality by reducing lipid and cholesterol levels [137,138,139].

### 3.3. Ruminants

Ruminant cattle production is essential for meeting the global protein demand (both meat and dairy products), but conventional feeding methods heavily rely on resource-intensive feed production, leading to environmental degradation [140]; additionally, ruminant livestock contributes 14.5% of greenhouse gas emissions [11,141,142]. Cattle release methane into the atmosphere by exhaling primarily through their mouths and nostrils. Methane is produced during enteric fermentation in the foregut of ruminants, where 95% is expelled through belching and the remaining 5% through the hindgut, expelled through the anus. It is important to highlight that the production of enteric methane represents a loss of dietary energy and constitutes an inefficiency in livestock feeding. Therefore, it is crucial to consider livestock feed ingredients to reduce methanogenesis, improve nutrition, and mitigate CH4 emissions [143]. Macroalgae may be a suitable alternative feed source due to their nutritional value; studies show that including macroalgae in ruminant diets improves animal performance and reduces methane emissions [144]. Sofyan et al. (2022) found varied effects of brown, green, and red macroalgae on methane production in ruminants [145] (Figure 2).

Notably, *Ascophyllum nodosum* and *Asparagopsis taxiformis* demonstrated significant potential for methane (CH_4_) reduction, although their effects on dairy cows and small ruminants were minimal. The study suggests that incorporating macroalgae into animal feed can effectively mitigate CH_4_ emissions without compromising animal performance [145]. Nevertheless, caution should be exercised regarding the elevated levels of bromoform and iodine residues in milk when utilizing high levels of *A. taxiformis*. Macroalgae contain bioactive compounds that can suppress methanogenesis and improve livestock health. *A. nodosum* promotes weight gain and enhances meat quality while reducing saturated fatty acids [126,148]. Including *A. taxiformis* and *Asparagopsis armata* in cattle and sheep feed can reduce methane production by up to 98% [146]. Roque et al. (2019) found that Holstein cows consuming *A. armata* showed a significant decrease in CH_4_ production of 26% at 0.5% inclusion and 67% at 1% inclusion, improving feed utilization efficiency [32]. Feed consumption was reduced by 10.8% and 38.0% for the respective inclusion levels, and CH_4_ yield was reduced by 20.3% and 42.7%. Also, Kinley et al. (2020) examined *A. taxiformis* as a feed component for the primary reduction of enteric CH_4_ in cattle. Feeding cows with *Asparagopsis* spp. reduces CH_4_ emissions by up to 98% and improves weight gain by up to 53%, with no adverse effects on feed consumption or meat quality [149]. In addition to the controversial efforts to reduce emissions by meat consumption, dairy products come from the same ruminal cattle.

Limited scientific evidence currently exists regarding the supplementation of ruminant diets with macroalgae and its impact on milk yield and composition. However, several studies have provided valuable insights on this topic. Studies have shown that feeding *Lithothamnion calcareum* [150] to dairy cows and supplementing lactating ewes’ diets with a combination of *A. nodosum* and flaxseed resulted in a higher milk production [151] and improved oxidative stability of fats. Additionally, cows supplemented with *A. nodosum* showed increased levels of δ-tocopherol in their milk [152]. A dietary supplement of pineapple oil, garlic, and brown algae in Holstein cows also demonstrated antioxidant activity, reduced COX-2 expression, and increased milk production; additionally, cows in the supplement group displayed a higher milk production and a tendency to engage in increased rumination when experiencing heat stress compared to the control group. Despite the potential benefits, challenges must be addressed to successfully integrate macroalgae into ruminant diets [104]. These include identifying suitable macroalgae species, optimizing the processing techniques, assessing the long-term effects on animal health and productivity, and the economic viability [12,153].

### 3.4. Pigs

Pork production plays a vital role in meeting the global demand for protein [154]. However, pig farming poses challenges related to environmental sustainability and animal health [155]. Since the middle of the last century, studies have been reported using macroalgae as a sustainable dietary supplement for pigs. However, a significant increase in interest in this area has been found since the year 2000. In addition to their role as nutritional supplements, macroalgae are known to be rich in beneficial bioactive compounds for pigs [156]. They offer a potential solution to reduce dependency on antimicrobials and antibiotics while serving as prebiotics to prevent gastrointestinal diseases, especially in weaned piglets [157,158]. All three types of macroalgae have been extensively studied as feed and nutraceuticals in pigs due to their observed ability to enhance the immune system, improve the oxidative stability of meat, and promote intestinal health [159,160,161]. Polysaccharides found in red algae, such as laminarin, fucoidan, ulvans, and carrageenan, are responsible for this. Among these, brown algae supplementation in pigs is particularly preferred as laminarin demonstrates anti-inflammatory properties, thereby mitigating the proinflammatory cytokine response [162,163]. Moreover, brown algae supplements may boost the immune system by stimulating immunoglobulin production and modulating cytokine production [28].

### 3.5. Other Animals

Macroalgae cultivation and commercialization are essential in fisheries and aquaculture economic sectors. Despite the potential benefits of macroalgae-based foods as sustainable and nutritious components in animal feed, a lack of scientific literature has hindered their widespread adoption by domesticated animals. Macroalgae can provide essential nutrients to domestic animals, including chelated micro-minerals that are more readily absorbed than inorganic minerals [164]. Complex carbohydrates with prebiotic properties, pigments, and polyunsaturated fatty acids found in macroalgae can also positively impact consumer health [165]. Therefore, this section provides a comprehensive scientific overview of recent studies conducted within the past five years. These studies investigate the potential advantages of incorporating macroalgae-based feeds to promote the health and growth of various domestic animal species, such as cats, dogs, and rabbits.

Despite the widespread use of *Ascophyllum nodosum*, a brown alga (Phaeophyceae) from the *Fucaceae* family, in animal products, there is a lack of studies assessing its impact on pet feed palatability. Isidori et al., in 2019, studied the effect of *A. nodosum* (0.3 & 1% of inclusion) on the palatability of extruded dog foods using the split plate test, and reported a negative impact [166]. Furthermore, a study by Pinna et al. (2021) examined the effects of supplementation with *Ascophyllum nodosum*, *Undaria pinnatifida*, *Saccharina japonica* (brown seaweed), and *Palmaria palmata* (red seaweed) in healthy adult dogs [167]. The study administered 15 g/kg per 28 days and evaluated parameters such as the fecal bacterial metabolites, fecal IgA, and tract digestibility of nutrients. However, no significant effects were observed. These findings suggest that further investigation is necessary regarding the formulation and processing of dog feed. While there are limited reports indicating that macroalgae may not be recommended for canine feeding, conducting more comprehensive tests is necessary to make informed decisions. Regarding cat feed, only one study has investigated the effects of dietary supplementation with enzymolysis algae powder (20 g/kg of food). The results showed that the supplemented group displayed an increase in Bacteroidetes, *Lachnospiraceae*, *Prevotellaceae*, and *Faecalibacterium*. This suggests that the addition of macroalgae powder to the diet may enhance gut health and gut microbiota [168]. On the other hand, a study by Abu Hafsa et al. (2021) reported that the inclusion of 1% *U. lactuca* in rabbit diets positively affected gut health and growth [169]. Additionally, other studies have indicated that including extracts of *Laminaria digitate*, such as polysaccharides and polyphenols, at concentrations of 0.3–0.6%, can enhance the antioxidant status and metabolism of fat while providing essential minerals and nutrients, thereby promoting animal growth [170,171,172]. Two similar studies employing the same percentage inclusion of macroalgae extracts reported improved growth, reduced cholesterol, enhanced oxidative stability, and an improved sensory quality of the meat [173,174]. Finally, in accordance with sustainable agriculture programs, specifically the European Green Deal Plan, incorporating seaweed in rabbit nutrition has been highlighted in a review paper as a promising approach to improve animal health. The integration of seaweed into rabbit nutrition serves as a nutraceutical, a viable alternative to antibiotics, and it can potentially enhance gut health [175].

## 4. Major Commercial Products from Macroalgae in the Feeding Market

Macroalgae cultivation and commercialization are part of the fisheries and aquaculture economic sector, a very important economic activity contributing to the growing domestic products (GDP). This sector accounts for about 0.2% of the GDP [176]. However, when analyzed per region, the highest contribution of this sector is reported for Oceania (excluding Australia and New Zealand), accounting for about 1.8% of the GDP. Current estimates value the macroalgal global market between USD 6.5 billion to USD 11.3 billion for 2021 and 2022, which is expected to increase at a 6–8.7% compound annual growth rate. In addition, increases in the production of macroalgae worldwide are also reported [177]. According to recent reports, global macroalgae cultivation increased from 10.6 million t (live weight) in 2000 to 35.1 million t in 2020, equivalent to a 231% increase [176]. The significant species contributing to this production are *L. japonica*, *Eucheuma* spp., *Gracilaria* spp., *U. pinnatifida*, *Porphyra* spp., *K. alvarezii*, *Sargassum fusiforme*, and *Eucheuma denticulatum*.

Besides cultivation, the formulation of products and their consumption in the feeding market is notorious. In Table 3, we presented macroalgae products commercially available in the market. It can be observed that most of the listed products correspond to cattle. This can be explained by the recent interest in feeding *Asparagopsis* sp. to cattle, which has shown a reduction in CH_4_ production above 80% [141]. Other commercial feed products of macroalgae include poultry, horse, and swine feed, as well as aquariums and aquaculture.

## 5. Future Perspectives

The growing demand for food, especially for proteins of animal origin, makes the livestock and aquaculture industries focus on the search for sustainable and environmentally friendly production alternatives [179]. Advances in technology and research are focused on developing new functional feeds for animals that not only help with nutrition and growth but also provide benefits for animal health [180]. Some studies offer optimistic prospects for integrating macroalgae as functional ingredients for animal feed production. Macroalgae can be sustainable protein sources for animal feed, replacing expensive ingredients, reducing the prices of animal products such as meat, milk, and eggs, and helping the economy of producers and consumers [181]. Likewise, the action of bioactive compounds from algae extracts can improve animal health and impact the quality of products of animal origin, being more beneficial for human consumption [20].

Future research should aim to determine the long-term effects of macroalgae on animal performance and the quality of protein products, including meat, milk, and eggs consumed by humans. Such investigations can potentially promote the scaling up of commercialization and the widespread adoption of macroalgae in feed production [182]. Nevertheless, understanding the specific requirements and preferences of various livestock farms is essential for deciding which species of macroalgae should undergo processing. Additionally, it is crucial to evaluate the biochemical and nutritional composition of macroalgae, which can vary based on factors such as species, geographical location, season, and external conditions [183]. These variations may result in different outcomes concerning animal health, growth, and product quality [184]. Therefore, comprehensive knowledge about the optimal sites for macroalgae growth, harvesting, and processing is imperative.

This review shows the positive impact of supplementing with different percentages of macroalgae in feeds for aquatic organisms, livestock, and pets. Palatability is a fundamental characteristic of the different rates of inclusion of macroalgae in animal feed [185]. Monogastric animals may reject meals with high percentages of macroalgae due to the sensory characteristics of the feed, such as the particle size, fracture strength, and dry matter content [186]. Different biomass processing techniques, such as salt extraction, drying, and milling, which can enhance the palatability in monogastric terrestrial animals [187], should be reviewed. Another technique to consider is the fermentation of macroalgae. This method has been shown to serve as a functional pretreatment of biomass, resulting in a fermentation process that produces appealing volatile compounds. Additionally, it can introduce lactic acid bacteria to act as animal probiotics and yield prebiotics through the fermentation of polysaccharides. This process also eliminates anti-nutritional compounds like phenols and tannins, which can negatively affect the palatability [188], also, it can negatively affect the sensory attributes of meat or milk [189]. It is essential to consider it in detail since the antinutrients such as alginate in macroalgae can affect its digestibility in monogastric animals [190].

On the other hand, bioactive compounds from macroalgae can be used to improve aspects, such as stimulation of the immune system, improved health, and enhanced antioxidant and antimicrobial properties. One disadvantage of bioactive compound extracts is that they require more processing, extraction equipment, and solvents [191]. More innovative procedures, like hydrothermal or microwave processing of seaweed, have been employed to enhance the bioavailability of nutrients such as proteins, pigments, and fatty acids. Simultaneously, these methods eliminate antinutrients such as polysaccharides that can negatively impact palatability (Magnusson). For the sustainable production and promotion of feed with veterinary properties aimed at enhancing animal health, there is a growing interest in utilizing macroalgae extracts, such as laminarin, fucoidan, and phlorotannin exclusively. Optimizing green extraction processes and establishing economically viable biorefineries is imperative to achieve this goal. Consequently, research efforts should be directed toward exploring large-scale extractions and assessing the long-term impacts on animal health and animal-derived protein products.

Related to the above, there are industrial limitations for the processing of macroalgae, and the formulation and preparation of animal feed since there is a lack of knowledge about the profitability and scaling of processing plants for this type of feed. Several factors must be considered to scale up and industrialize this promising feed technology. Firstly, significant amounts of algal biomass are necessary to ensure the viability of the products in the market. The feed industry should engage in large-scale macroalgae cultivation through aquaculture or by utilizing invasive algae species like sargassum. This approach aligns with the principles of a circular economy. Also, macroalgae farming can be a strategy to mitigate greenhouse gases, treat agro-industrial effluents, and reduce carbon footprint [192]. Therefore, it is imperative to conduct studies on the carbon sequestration-based life cycle assessment of macroalgae, from cultivation to feed production and consumption, to assess the balance of carbon sequestration by algal culture and the total carbon release during the life cycle of feed production [193,194].

## 6. Conclusions

In this review, we found that macroalgae or their extracts emerge as an alternative supplement for the formulation of feed based on their nutritional profile and bioactive compound concentration. Despite its potential for feed, challenges persist. These include the need for acceptance, palatability, inclusion level, and conducting studies focusing on the immunological or digestive response in terrestrial animals, specifically monogastric organisms. Due to their aquatic origin, macroalgae are well-accepted within aquaculture. However, there is a need to address improvements in feed formulation technology to counter diseases and ensure sustainable production. Moreover, further research is imperative to understand the long-term effects of incorporating macroalgae into feeds, particularly concerning the quality of meat and animal products such as milk and eggs. The challenge lies in effectively integrating macroalgae through interdisciplinary approaches, contributing to improving aquaculture and livestock practices. Such efforts will expand the commercialization of macroalgae-based feed globally, aligning a circular economy approach to combat climate change.

## Figures and Tables

**Figure 1 plants-12-03609-f001:**
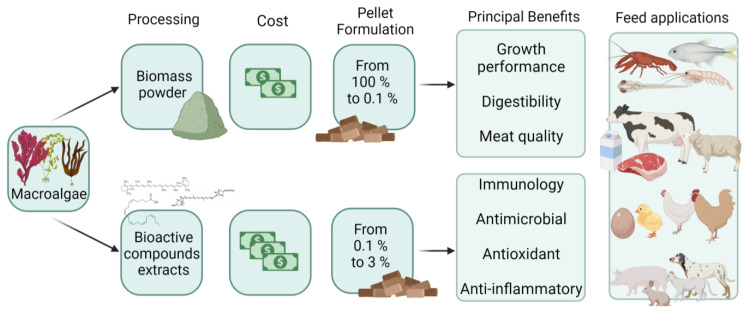
General diagram of the use of macroalgae for animal feed and their nutritional benefits.

**Figure 2 plants-12-03609-f002:**
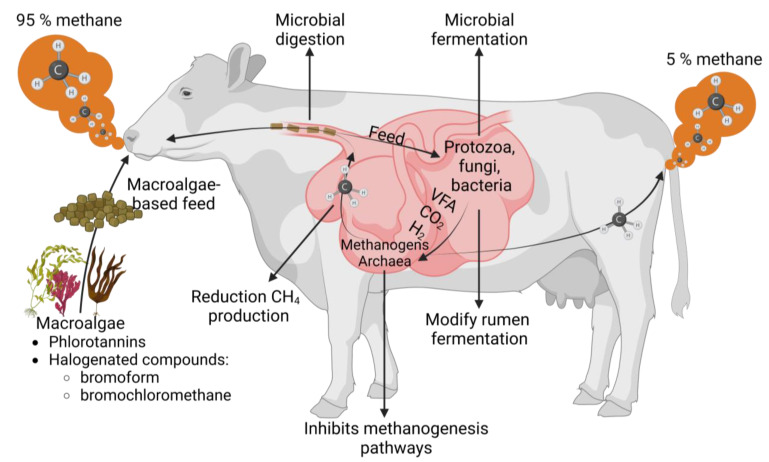
Effect of macroalgae intake on methane gas production by ruminant fermentation. Notes: methane emissions from burps represent 95% and flatulence the 5% of the total emitted by livestock [146,147].

**Table 1 plants-12-03609-t001:** List of macroalgae sources, bioactive molecules and compounds, and their applications.

Seaweed	Group	Compound	Source	Use	Bioactivity Reported	Reference
Brown seaweed (Phaeophyta)						
Brown	Polyphenol	Phlorotannins	*Ascophhyllum nodosum*, *Eisenia arborea*, *Fucus* spp., and *Sargassum* spp.	Modulate gut microbiota.	Antimicrobial and bacterial inhibitor. Anti-inflammatory effects.	[51,52,53]
Brown	Polysaccharide	Alginate	*Phaeophyceae* spp.	Numerous applications in food and biotechnology. Gelling abilities, stabilizing, water holding capability.	Antioxidant, Anti-inflammatory, antimicrobial, and antitumor properties. Growth-promoting effects in plants and animals.	[54,55,56]
Brown	Polysaccharide	Fucoidan	*Cladosiphon novae* *caledoniae*, *Fucus vesiculosus*, *Hizikia fusiformis*, *Phaeophyceae* spp., *Saccharina japonica*, *Sargassum crassifolium*, and *Undaria pinnatifida*	Functional foods and nutraceuticals. Prebiotic properties.	Antioxidant, anti-inflammatory, antiangiogenic, anticoagulant, immunomodulatory, anti-adhesive, antitumoral, antidiabetic, antimicrobial, and anti-neurodegenerative.	[57,58]
Brown	Polysaccharide	Laminarin	*Laminaria* spp.	Source of arabinose, galactose, mannose, xylose, glucose, rhamnose, and glucuronic acid.	Antitumour, antihyperglycaemic, and antihyperlipidaemic agents (mouse model).	[9,59]
Red seaweed (Rhodophyta)						
Red	Polysaccharide	Agar	*Agarophyton vermiculophyllum*, *Gelidium* spp.	Gelling thickener and clarifying agent.	Enhance bifidobacterial populations (in vitro). Increase complement pathway activity in fish (*Pangasius bocourti*).	[1,60,61]
Red	Polysaccharide	Carrageenan	*Furcellariaceae* spp., *Gigartinaceae* spp., *Hypneaceae* spp., *Phyllophoraceae* spp., and *Solieriaceae* spp.	Gelling, thickening, emulsifying and stabilizing agents. These are the functional ingredients commonly found in vegetable-based products, dairy, baked items, meat, and fish.	Antioxidant, antiviral, antibacterial, antihyperlipidemic, anticoagulant, anticancer, and immunomodulatory effects.	[49,62,63]
Red	Protein	Allophycocyanin, Phycocyanin, Phycoerythrin, and Phycoerythrocyanins	*C. crispus*, *G. gracilis*, *G. turuturu*, *Gelidium amansii. Porphyra* spp., *Porphyridium cruentum*, and *Rhodosorus marinus*	Colorants for food and cosmetics Immune system improvement.	Antioxidant, antitumoral, antidiabetic, anti-inflammatory, antioxidative, and anti-irradiative effects.	[64,65,66,67]
Green seaweed (Chlorophyta)						
Green	Polysaccharide	Ulvan	*Entermorpha* sp. and *Ulva* sp.	Gelling and rheological properties like gum arabic.	Antibacterial antitumor, antioxidant, anti-thrombolytic, immunomodulation, antiviral, and anticoagulant activity, pharmacokinetics, oxidative and thermal stability	[57,62,68]

**Table 2 plants-12-03609-t002:** The effects of dietary seaweeds on growth performance, physiology, and immune response in farmed crustaceans.

Seaweed Specie	Used Compound	Test Organism	Trial Dose Days	Pathogen/ Stress	Results	Reference
Brown seaweed (Phaeophyta)						
*Cystoseira* *trinodis*	Polysaccharide—fucoidan	*Litopenaeus vannamei*	0, 0.1, 0.2 and 0.4%—60 days	WSSV	FW, WG, SGR, expression rate of genes: proPO I, SOD, LYZ, and resistance against WSSV +; GPx and FCR −	[108]
*Sargassum* *horneri*	Hot water extract	*L. vannamei*	0, 2.5, 5.0, 10 g kg^−1^—28 days	NA	PO, THC, phagocytic rate, WG, and expression rate of genes: ProPO I, ProPO II, peroxinectin, α2macroglobulin, clotting protein, LYZ, SOD, GPx, penaiedin2-4, crusting +	[109]
*U. pinnatifida*	This is a hot water extract that contains a significant amount of mannitol and fucoidan	*Pleoticus muelleri*	0, 3.0 g 100 g^−1^—30 days	UVR stress	Resistance against UVR stress, concentrations of UV-absorbing compounds, and TAS +; carotenoid concentration −	[110]
*Iyengaria* *stellata*	Water extract	*L. vannamei*	0, 0.5, 1, 1.5 g kg^−1^—56 days	*Photobacterium damselae*	WG, SGR, FE, PUFA, TAS, PO, CAT, SOD and GP_X_ activities, and resistance against *P. damselae* +	[99]
*Padina tetrastromatica* and *Sargassum ilicifolium*	Ethyl acetate, ethanol, and methanol extracts	*Penaeus* *monodon*	0, 2.5, and 5 g kg^−1^—45 days	*Vibrio parahaemolyticus*	WG, SGR, SR, antibacterial activity, PO, SOD, resistance against *V. parahaemolyticus +*; minimal time required for hemolymph clotting −	[111]
*S. cristaefolium*	Wholemeal	*L. vannamei*	0, 5, 10, 15, 20 and 40 g kg^−1^—60 days	NA	FW, muscle total protein =; muscle cholesterol and triglyceride level, and *Vibrio* counts in the intestine −; THC +	[112]
*Sargassum* *polycystum*	Powdered seaweed flour	*L. vannamei*	0.5 g kg^−1^—56 days	Cold stress	Body proximate composition: protein, ether extract, ash, carcass energy, FCR −; WG, ADG, SGR, SR, nonspecific immune responses, activation of the hepatic glandular duct system, hemocyte infiltration, and expression rate of genes: SOD, penaeidin4, HSP-70 +	[113]
Red seaweed (Rhodophyta)						
*Amphiroa* *fragilissima*	Crude polysaccharides, encapsulated *Artemia nauplii*	*L. vannamei*	0, 0.1, 0.15, and 0.20 g L^−1^—45 days	NA	WG, SGR, protease/amylase activities, and: protein, amino acid, free sugar, lipid, SOD, and CAT activities +	[114]
*Asparagopsis* *armata*	Ethanolic extract	*L. vannamei*	0, 1.5, 3.5, 7.5 g kg^−1^—40 days	*V. parahaemolyticus*	WG, SR, TAS, antimicrobial activity, and resistance against *V. parahaemolyticus +*; FCR −	[102]
*Gracilaria* *birdiae*	Sulfated polysaccharide	*L. vannamei*	0, 0.3%—32 days	WSSV	Agglutinating capacity, PO, resistance against WSSV, and THC +	[100]
*Gracilaria* *verrucosa*	Ethyl acetate extracts (alkaloids, saponins, phenolics, flavonoids, triterpenoids, steroids, and glycosides)	*L. vannamei*	0—2.0 g per kg^−1^—14 days	*Vibrio harveyi*	Inhibition the growth of *V. harveyi*, THC, PO activity, phagocytosis activity, respiratory burst activity, and resistance against *V. harveyi +*	[107]
*Porphyra* *haitanensis*	Wholemeal	*L. vannamei*	0, 2.0%—56 days	WSSV	FCR -; WG, SGR, digestibility, TAS, expression of immune genes, ProPO and SOD activities +	[105]
*Jania* *adherens*	Ethanolic extract	*L. vannamei*	0, 0.5, 1.0, and 1.5 g kg^−1^—56 days	*Photobacterium damselae*	FCR −; SR =; WG, SGR, FE, PO, GPx, lipase, amylase, LYZ, respiratory burst activities, and resistance against *P. damselae +*	[99]
*K. alvarezii*	Fermented *K. alvarezii* powder	*L. vannamei*	0, 0.5, 1.5%—15 days	NA	FW, SGR, and SR +	[115]
*Kappaphycus* *alvarezii*	κ-carrageenan	*P. monodon*	0.0, 0.15, 0.30, 0.45, and 0.60 g kg^−1^—30 days	Salinity stress	Attractability, FW, WG, SGR, FE, and immune-stimulating effects on salinity stress +; FCR −	[103]
*Sarcodia* *suae*	*S. suae* powder	*L. vannamei*	0, 2.5, 5, 7.5%—20 days	*Vibrio alginolyticus*	Anti-*V. alginolyticus* activity, phagocytic activity, THC, and expression of GPx +	[116]
Green seaweed (Chlorophyta)						
*Caulerpa* *lentillifera*	Seaweed flour	*P. monodon*	0, 10, 20, 30, 40 and 50 g kg^−1^—60 days	NA	WG, SGR, and FR+; FCR −; SR =	[117]
*Caulerpa* sp.	*Caulerpa* sp. flour	White leg shrimp	0, 2, 4 and 6 g kg^−1^—30 days	NA	FW and FE +; FCR −	[118]
*Enteromorpha*	Polysaccharides	*Fenneropenaeus* *merguiensis*	0, 1, 2 and 3 g kg^−1^—42 days	NA	FCR, relative abundance of *Vibrio* spp., and malondialdehyde content −; expression of immune genes, relative abundance of Firmicutes, FW, WG, SGR, TAS, GPx, S-transferase, SOD, LYZ, and PO activities +	[119]
*U. lactuca*	Water extract	*L. vannamei*	0, 5, 10 15%—28 days	NA	FW, WG, SGR, FE, chymotrypsin, lipase, and amylase enzyme activity +	[120]
*U. lactuca*	*U. lactuca* powder	*L. vannamei*	0, 4 g 100 g^−1^—36 days	NA	SGR, ADG, and relative abundance of: *Agarivorans, Sphingomonas, Lactobacillus, Leuconostoc, Peredibacter, Bdellovibrios* +; relative abundance of: *V. alginolyticus* and *Photobacterium* sp. −	[121]
*Ulva clathrata* and *U. lactuca*	Wholemeal	*L. vannamei*	100%—28 days	NA	Relative abundance of *Rubritalea*, *Lysinibacillus*, *Acinetobacter*, *Blastopirellula* and *Litoreibacter* spp. +; FW, SR =; relative abundance of *Vibrio* −	[122]
*Ulva* *intestinalis*	Hot water crude extract	*L. vannamei*	0, 1, 5, and 10 g kg^−1^—28 days	WSSV and YHV	WG, ADG, SGR, villus height, phagocytic activity, resistance against YHV, and expression of immune genes +; resistance against WSSV =	[123]
*Ulva* *prolifera*	Polysaccharides and filtered residue	*L. vannamei*	0, 0.78, 1.33, 31.7 g kg^−1^—21 days	*V. parahaemolyticus*	FCR −; FW, LYZ, PO, resistance against *V. parahaemolyticus*, and immune protective rate +	[124]
Combined seaweed						
*Sargassum filipendula* and *U. pinnatifida*	Combined seaweed dry biomass	*L. vannamei*	0, 15, 25, 45 g kg^−1^—15 and 21 days	Thermal shock/WSSV	Resistance against WSSV, hemocyte infiltration, and positive changes in shrimp gut microbiome +	[125]
*S. filipendula* and *U. pinnatifida*	Combined seaweed dry biomass	*L. vannamei*	0, 25, 30, 45, 50 g kg^−1^—15 and 49 days	Thermal shock/*Vibrio* spp.	Resistance against *Vibrio* spp. and against thermal shock FW, and heterotrophic bacteria count in organisms +	[126]
*U. lactuca and* *Jania rubens*	Combined seaweed powder	*Procambarus clarkii*	0, 10, 15%—56 days	NA	Frequency of molting and FW+; FCR −; SR =	[127]
*U. pinnatifida* and *S. filipendula*	Combined seaweed dry biomass	*L. vannamei*	0, 3 and 5%—35 days	Thermal stress/WSSV	Gut bacterial diversity, THC, and resistance against thermal stress +; abundance of *Vibrionaceae* spp. −; SR, FCR, and FW =	[101]
*U. lactuca*, *Eisenia* sp., and *Porphyra* sp.	Mix in equal parts of extracts	*L. vannamei*	0.30 g kg^−1^—28 days	WSSV	Resistance against WSSV (genes related to WSSV resistance) +	[128]

ADG: Average daily gain, FCR: feed conversion ratio, FE: feed efficiency, FW: organism final weight, GPx: glutathione peroxidase, HSP: heat shock protein, LYZ: lysozyme, NA: not applicable, PO: phenoloxidase, ProPO: prophenoloxidase, SGR: specific growth rate, SOD: superoxide dismutase, SR: survival rate, TAS: total antioxidant, THC: total hemocytes count, WG: weight gain, WSSV: white spot syndrome virus, YHV: yellowhead virus, +: positive change, −: negative change, =: no change.

**Table 3 plants-12-03609-t003:** Macroalgae products commercially available in the market for animal feeding.

Product Name	Macroalgae	Application	Country	Reference
A la Carte	*Palmaria palmata*, *Porphyra umbilicalis*, *Porphyra yezoensis*	Seawater fish—aquarium	France	https://www.aquariumsystems.eu/ accessed on 19 May 2023
AlgaeMAX	*Ulva*, *U. pinnatifida* (wakame), Kelp	Seawater fish—aquarium	United States	https://www.feedspectrum.com/seaweeds accessed on 19 May 2023
Algimun	Not specified (algal extracts)	Cattle, poultry, swine, horse, fish	France	https://www.olmix.com/animal-care/algimun accessed on 22 May 2023
AlgoFeed+	Not specified (algal extracts)	Swine, poultry, and fish	France	https://www.olmix.com/animal-care/algofeedtm accessed on 22 May 2023
AquaArom	*Laminaria* sp.	Salmonid feed	Canada	https://addican.com/aquaarom/ accessed on 19 May 2023; [178]
Betterplants seaweed meal	*A. nodosum*	Cattle and horses	Ireland	https://betterplants.ie/product-category/natural-animal-nutrition/ accessed on 19 May 2023
Brominata	*A. taxiformis*	Cattle	United States	https://blueoceanbarns.com/ accessed on 19 May 2023
FutureFeed	*Asparagopsis* sp.	Ruminants	Australia	https://www.future-feed.com accessed on 19 May 2023
Garnelio Seealgenmehl	Not specified	Fan shrimp and mussels	Germany	https://www.garnelio.de/en/garnelio-seaweed-flour-25-g accessed on 19 May 2023
NAF Life-Guard Seaweed	Not specified	Poultry	United Kingdom	https://www.naf-equine.eu/uk/life-guard/lg-seaweed accessed on 19 May 2023
Nettex Poultry Seaweed	Not specified	Poultry	United Kingdom	https://www.nettexpoultry.com/ accessed on 19 May 2023
OceanFeed	Not specified	Bovine, equine, swine, poultry, aqua	United Kingdom	https://oceanharvesttechnology.com/ accessed on 24 May 2023
Plankton vital	Nori	Freshwater and seawater fish—aquarium	Germany	https://planktonvital.de/ accessed on 19 May 2023
Sea Forest	*Asparagopsis* sp.	Cattle and sheep	Australia	https://www.seaforest.com.au/ accessed on 19 May 2023
SeaGraze	*A. taxiformis*	Ruminants	United States	https://symbrosia.co/seagraze accessed on 19 May 2023
Seaperia Seaweed Meal	*A. nodosum*	Ruminants	Australia	https://www.seaperia.com/ accessed on 19 May 2023
Seaweed extreme	Nori	Tropical seawater fish—aquarium	Japan	https://www.hikari.info/product/index.html accessed on 19 May 2023
Seealgenmehl	*A. nodosum*	Horse	Germany	https://www.pernaturam.de accessed on 19 May 2023
ShiLai	*A. taxiformis*	Cattle	China	https://asparagopsistaxiformisfeed.com/ accessed on 19 May 2023
Volta Seaweed	*A. taxiformis*	Cattle	Sweden	https://www.voltagreentech.com/ accessed on 19 May 2023

## Data Availability

Not applicable.

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
