# Peer review of "New Perspective for Macroalgae-Based Animal Feeding in the Context of Challenging Sustainable Food Production"

_plants, 2023, doi:10.3390/plants12203609_

Round 1

Reviewer 1 Report

  In this work, the authors gather information on the possibility of using macroalgae biomass for animal feed, taking advantage of its many bioactive and nutritious components. This review is very comprehensive, and the subject is relevant and industrially important in the context of moving towards sustainable and ecological processes. The bibliographic references are appropriate and very recent (most of them from 2013 to 2023). 

Please, address also the following comments/suggestions/questions.

1 - Correct, throughout the text and figures: “meters” to m, “metric tons” to “t”.

2 – In the tables, list the macroalgae sp names by alphabetical order; in Table 3, list the Product name by alphabetical order.

3 – In the tables, full stops should be removed.

4 – Correct, in Table 1 :“Improves the immune system.” to “Immune system improvement”).

5 – In line 242, correct “The Table 2 provides”… to “Table 2 provides”…

6 – In Table 2, adjust the column width for “Trial Doses – Days”, to facilitate the reading.

7 – In Figure 2, it is not clear what “95% methane” and “5% methane” is. A % reduction when using macroalgae? A % composition? The % excreted through cow belching or cow flatulence? Is it relevant in this manuscript? If so, please modify it, as it is not clear even after reading the text related to this figure. In fact, these values are never referred in the text.

8 – Line 342 Please, rewrite the sentence “In some cultures, pigs are treated as pets but, in most, as human food, more studies have been found on diets based on macroalgae than on the previously mentioned pets.” Its meaning is not clear.

9 – Line 491 “funding acquisition, R.P.S.” Line 493 “Funding: not applicable”; this is contradictory; the authors should refer the funding or change the author’s contribution.

10 -  Line 495  correct “Acknowledgments” to “Acknowledgements”.

In spite of its relevance, in my opinion, this manuscripts needs to be reviewed to improve the English language. Many sentences are difficult to understand and should be rewritten (2 examples in the Abstract, but there are a few more throughout the text: (i) “They are classified into three categories: green (Chlorophyceae), red (Rhodophyceae), and brown (Phaeophyceae), and are commonly used as part of aquafeed, but due to the demand for food in the market.” (????); (ii) “Considering seaweed have a high growth rate and do not require arable land, are a rich source of proteins, essential amino acids, carbohydrates, minerals, and vitamins, as well as biologically active compounds such as polyunsaturated fatty acids, antioxidants (polyphenols), and pigments (chlorophylls, carotenoids) with beneficial properties such as antibacterial, antifungal, antiviral, antioxidant, and anti-inflammatory, and they can be used as prebiotics.” (????))

Author Response

Reviewer 1

In this work, the authors gather information on the possibility of using macroalgae biomass for animal feed, taking advantage of its many bioactive and nutritious components. This review is very comprehensive, and the subject is relevant and industrially important in the context of moving towards sustainable and ecological processes. The bibliographic references are appropriate and very recent (most of them from 2013 to 2023). 

Dear reviewer, we appreciate your comments to improve our manuscript. We have addressed this accordingly  and the changes were highlighted in GREEN COLOR through the Manuscript.

Please, address also the following comments/suggestions/questions.

1 - Correct, throughout the text and figures: “meters” to m, “metric tons” to “t”.

Authors response: Thank you for your comments, the changes have been addressed.

2 – In the tables, list the macroalgae sp names by alphabetical order; in Table 3, list the Product name by alphabetical order.

Authors response: Thank you for your comment. The table have been restructured and reorganized.

3 – In the tables, full stops should be removed.

Authors response: Thank you for your recommendation. The changes have been applied.

4 – Correct, in Table 1 :“Improves the immune system.” to “Immune system improvement”).

Authors response: Thank you for your observation. The change has done.

5 – In line 242, correct “The Table 2 provides”… to “Table 2 provides”…

Authors response: Thank you for your observation. The change has done.

6 – In Table 2, adjust the column width for “Trial Doses – Days”, to facilitate the reading.

Authors response: Thank you for your observation. The change has done.

7 – In Figure 2, it is not clear what “95% methane” and “5% methane” is. A % reduction when using macroalgae? A % composition? The % excreted through cow belching or cow flatulence? Is it relevant in this manuscript? If so, please modify it, as it is not clear even after reading the text related to this figure. In fact, these values are never referred in the text.

Authors response:

Thank you for your feedback. We include notes and additional information related to the percentages. In the manuscript, we have elaborated on the significance of these values, specifying that they represent the percentage of methane excreted through cow belching (95%) and cow flatulence (5%) to provide a more comprehensive explanation.

8 – Line 342 Please, rewrite the sentence “In some cultures, pigs are treated as pets but, in most, as human food, more studies have been found on diets based on macroalgae than on the previously mentioned pets.” Its meaning is not clear.

Authors response: Thank you for your comment. In order to improve the reader's understanding, the aforementioned line has been removed.

9 – Line 491 “funding acquisition, R.P.S.” Line 493 “Funding: not applicable”; this is contradictory; the authors should refer the funding or change the author’s contribution.

Authors response: Thank you for your observation, such information has been added.

10 -  Line 495  correct “Acknowledgments” to “Acknowledgements”.

Authors response: Dear reviewer, technically, both the spellings are correct. It is spelled 'acknowledgement' in British English and 'acknowledgment' in American English. We will keep the word 'acknowledgment' because it is predefined in the journal's template.

In spite of its relevance, in my opinion, this manuscript needs to be reviewed to improve the English language. Many sentences are difficult to understand and should be rewritten (2 examples in the Abstract, but there are a few more throughout the text: (i) “They are classified into three categories: green (Chlorophyceae), red (Rhodophyceae), and brown (Phaeophyceae), and are commonly used as part of aquafeed, but due to the demand for food in the market.” (????); (ii) “Considering seaweed have a high growth rate and do not require arable land, are a rich source of proteins, essential amino acids, carbohydrates, minerals, and vitamins, as well as biologically active compounds such as polyunsaturated fatty acids, antioxidants (polyphenols), and pigments (chlorophylls, carotenoids) with beneficial properties such as antibacterial, antifungal, antiviral, antioxidant, and anti-inflammatory, and they can be used as prebiotics.” (????))

Authors response: Thank you for your comment. The language was revised by a native speaker and mistakes in the abstract were corrected.

Reviewer 2 Report

This paper provides a comprehensive review of the potential of macroalgae as a sustainable feed ingredient. It effectively summarizes the current knowledge on the topic, highlighting the benefits and challenges associated with incorporating macroalgae into animal feed. The review is well-structured and provides valuable insights for researchers and practitioners in the field. Overall, the review should address these issues to provide a comprehensive analysis of the benefits and challenges associated with incorporating macroalgae into animal feed.

-Acceptance and Palatability: The paper should address the challenges related to the acceptance and palatability of macroalgae-based feed in terrestrial animals, particularly monogastric organisms. This includes conducting studies on the immunological or digestive response to ensure the feasibility of incorporating macroalgae into their diets.

-Improvements in Feed Formulation Technology: The review should highlight the need for advancements in feed formulation technology to counter diseases and ensure sustainable production. This would contribute to the overall success and commercialization of macroalgae-based feed.

-Long-Term Effects on Meat and Animal Products: The paper should emphasize the importance of conducting further research to understand the long-term effects of incorporating macroalgae into animal diets, specifically regarding the quality of meat and animal products such as milk and eggs.

Author Response

Reviewer 2

This paper provides a comprehensive review of the potential of macroalgae as a sustainable feed ingredient. It effectively summarizes the current knowledge on the topic, highlighting the benefits and challenges associated with incorporating macroalgae into animal feed. The review is well-structured and provides valuable insights for researchers and practitioners in the field. Overall, the review should address these issues to provide a comprehensive analysis of the benefits and challenges associated with incorporating macroalgae into animal feed.

Dear reviewer, we appreciate your comments to improve our manuscript. We have addressed accordingly and the changes were highlighted in GREEN COLOR through the Manuscript

-Acceptance and Palatability: The paper should address the challenges related to the acceptance and palatability of macroalgae-based feed in terrestrial animals, particularly monogastric organisms. This includes conducting studies on the immunological or digestive response to ensure the feasibility of incorporating macroalgae into their diets.

Authors response: Thank you for your valuable feedback. We considered this suggestion and incorporated information about the challenges and potential solutions, such as fermentation and bioactive compound extraction in line 546.

-Improvements in Feed Formulation Technology: The review should highlight the need for advancements in feed formulation technology to counter diseases and ensure sustainable production. This would contribute to the overall success and commercialization of macroalgae-based feed.

Authors response: Thank you for your observation. We added information regarding new technologies for food processing based on macroalgae, focusing on obtaining bioactive compounds that support animal health and sustainable production in line 529.

-Long-Term Effects on Meat and Animal Products: The paper should emphasize the importance of conducting further research to understand the long-term effects of incorporating macroalgae into animal diets, specifically regarding the quality of meat and animal products such as milk and eggs.

Authors response: Thank you for your valuable comment. We have extended the information regarding this topic. Please see lines 529-539.

Reviewer 3 Report

Dear Authors, the manuscript is very interesting and macroalgae (e.g. Ulva, Palmaria palmata, etc.) can be food for humans or animal feed. However, the authors must also highlight critical factors such as the absorption of pollutants increasingly present in the seas such as PAHs, heavy metals, etc. by macroalgae. Furthermore, massive macroalgae farming can lead to a heavy reduction in dissolved oxygen, leading to dangerous alterations in the coastal ecosystem if the farming takes place below the coast. These aspects must be appropriately addressed by the authors. Best wishes  

Author Response

Dear Authors, the manuscript is very interesting and macroalgae (e.g. Ulva, Palmaria palmata, etc.) can be food for humans or animal feed. However, the authors must also highlight critical factors such as the absorption of pollutants increasingly present in the seas such as PAHs, heavy metals, etc. by macroalgae. Furthermore, massive macroalgae farming can lead to a heavy reduction in dissolved oxygen, leading to dangerous alterations in the coastal ecosystem if the farming takes place below the coast. These aspects must be appropriately addressed by the authors. Best wishes

Author´s response: 

Dear reviewer

Please find in the manuscript the detailed responses to your comments. All changes/corrections made have been addressed and highlighted with a YELLOW background in color through the Manuscript. Please see lines 126 to 148 in section 2.